# Combining Imitation and Reinforcement Learning with Free Energy Principle

## Abstract

Imitation Learning (IL) and Reinforcement Learning (RL) from high dimensional sensory inputs are often introduced as separate problems, but a more realistic problem setting is how to merge the techniques so that the agent can reduce exploration costs by partially imitating experts at the same time it maximizes its return. Even when the experts are suboptimal (e.g. Experts learned halfway with other RL methods or human-crafted experts), it is expected that the agent outperforms the suboptimal experts' performance. In this paper, we propose to address the issue by using and theoretically extending Free Energy Principle, a unified brain theory that explains perception, action and model learning in a Bayesian probabilistic way. We find that both IL and RL can be achieved based on the same free energy objective function. Our results show that our approach is promising in visual control tasks especially with sparse-reward environments.

## 1 Introduction

Imitation Learning (IL) is a framework to learn a policy to mimic expert trajectories. As the expert specifies model behaviors, there is no need to do exploration or to design complex reward functions. Reinforcement Learning (RL) does not have these features, so RL agents have no clue to realize desired behaviors in sparse-reward settings and even when RL succeeds in reward maximization, the policy does not necessarily achieve behaviors that the reward designer has expected. The key drawbacks of IL are that the policy never exceeds the suboptimal expert performance and that the policy is vulnerable to distributional shift. Meanwhile, RL can achieve super-human performance and has potentials to transfer the policy to new tasks. As real-world applications often needs high sample efficiency and little preparation (rough rewards and suboptimal experts), it is important to find a way to effectively combine IL and RL.

When the sensory inputs are high-dimensional images as in the real world, behavior learning such as IL and RL would be difficult without representation or model learning. Free Energy Principle (FEP), a unified brain theory in computational neuroscience that explains perception, action and model learning in a Bayesian probabilistic way (Friston et al., 2006; Friston, 2010), can handle behavior learning and model learning at the same time. In FEP, the brain has a generative model of the world and computes a mathematical amount called Free Energy using the model prediction and sensory inputs to the brain. By minimizing the Free Energy, the brain achieves model learning and behavior learning. Prior work about FEP only dealt with limited situations where a part of the generative model is given and the task is very low dimensional. As there are a lot in common between FEP and variational inference in machine learning, recent advancements in deep learning and latent variable models could be applied to scale up FEP agents to be compatible with high dimensional tasks.

Recent work in model-based reinforcement learning succeeds in latent planning from high-dimensional image inputs by incorporating latent dynamics models. Behaviors can be derived either by imagined-reward maximization (Ha & Schmidhuber, 2018; Hafner et al., 2019a) or by online planning (Hafner et al., 2019b). Although solving high dimensional visual control tasks with model-based methods is becoming feasible, prior methods have never tried to combine with imitation.

In this paper, we propose Deep Free Energy Network (FENet), an agent that combines the advantages of IL and RL so that the policy roughly learns from suboptimal expert data without the need of exploration or detailed reward crafting in the first place, then learns from sparsely specified reward functions to exceed the suboptimal expert performance.

The key contributions of this work are summarized as follows:

- **Extension of Free Energy Principle:**
  We theoretically extend Free Energy Principle, introducing policy prior and policy posterior to combine IL and RL. We implement the proposed method on top of Recurrent State Space Model (Hafner et al., 2019b), a latent dynamics model with both deterministic and stochastic components.

- **Visual control tasks in realistic problem settings:**
  We solve Cheetah-run, Walker-walk, and Quadruped-walk tasks from DeepMind Control Suite (Tassa et al., 2018). We do not only use the default problem settings, we also set up problems with sparse rewards and with suboptimal experts. We demonstrate that our agent outperforms model-based RL using Recurrent State Space Model in sparse-reward settings. We also show that our agent can achieve higher returns than Behavioral Cloning (IL) with suboptimal experts.

## 2 BACKGROUNDS ON FREE ENERGY PRINCIPLE

### 2.1 PROBLEM SETUPS

We formulate visual control as a partially observable Markov decision process (POMDP) with discrete time steps $t$, observations $o_t$, hidden states $s_t$, continuous action vectors $a_t$, and scalar rewards $r_t$. The goal is to develop an agent that maximizes expected return $\mathbb{E}[\sum_{t=1}^{T} r_t]$.

### 2.2 FREE ENERGY PRINCIPLE

Perception, action and model learning are all achieved by minimizing the same objective function, Free Energy (Friston et al., 2006; Friston, 2010). In FEP, the agent is equipped with a generative model of the world, using a prior $p(s_t)$ and a likelihood $p(o_t|s_t)$.

$$p(o_t, s_t) = p(o_t|s_t)p(s_t) \tag{1}$$

**Perceptual Inference**     Under the generative model, the posterior probability of hidden states given observations is calculated with Bayes' theorem as follows.

$$p(s_t|o_t) = \frac{p(o_t|s_t)p(s_t)}{p(o_t)}, \quad p(o_t) = \int p(o_t|s_t)p(s_t)ds \tag{2}$$

Since we cannot compute $p(o_t)$ due to the integral, we think of approximating $p(s_t|o_t)$ with a variational posterior $q(s_t)$ by minimizing KL divergence $KL(q(s_t)||p(s_t|o_t))$.

$$KL(q(s_t)||p(s_t|o_t)) = \ln p(o_t) + KL(q(s_t)||p(o_t, s_t)) \tag{3}$$

$$F_t = KL(q(s_t)||p(o_t, s_t)) \tag{4}$$

We define the Free Energy as (eq.4). Since $p(o_t)$ does not depend on $s_t$, we can minimize (eq.3) w.r.t. the parameters of the variational posterior by minimizing the Free Energy. Thus, the agent can infer the hidden states of the observations by minimizing $F_t$. This process is called 'perceptual inference' in FEP.

**Perceptual Learning**     Free Energy is the same amount as negative Evidence Lower Bound (ELBO) in variational inference often seen in machine learning as follows.

$$p(o_t) \geq -F_t \tag{5}$$

By minimizing $F_t$ w.r.t. the parameters of the prior and the likelihood, the generative model learns to best explain the observations. This process is called 'perceptual learning' in FEP.

**Active Inference**     We can assume that the prior is conditioned on the hidden states and actions at the previous time step as follows.

$$p(s_t) = p(s_t|s_{t-1}, a_{t-1}) \tag{6}$$

The agent can change the future by choosing actions. Suppose the agent chooses $a_t$ when it is at $s_t$, the prior can predict the next hidden state $s_{t+1}$. Thus, we can think of the Expected Free Energy $G_{t+1}$ at the next time step $t+1$ as follows (Friston et al., 2015).

$$
\begin{aligned}
G_{t+1} &= KL(q(s_{t+1})||p(o_{t+1}, s_{t+1})) = \mathbb{E}_{q(s_{t+1})}[\ln q(s_{t+1}) - \ln p(o_{t+1}, s_{t+1})] \\
&= \mathbb{E}_{q(s_{t+1})p(o_{t+1}|s_{t+1})}[\ln q(s_{t+1}) - \ln p(o_{t+1}, s_{t+1})] \qquad (7) \\
&= \mathbb{E}_{q(s_{t+1})p(o_{t+1}|s_{t+1})}[\ln q(s_{t+1}) - \ln p(s_{t+1}|o_{t+1}) - \ln p(o_{t+1})] \\
&\approx \mathbb{E}_{q(o_{t+1}, s_{t+1})}[\ln q(s_{t+1}) - \ln q(s_{t+1}|o_{t+1}) - \ln p(o_{t+1})] \qquad (8) \\
&= \mathbb{E}_{q(o_{t+1})}[-KL(q(s_{t+1}|o_{t+1})||q(s_{t+1})) - \ln p(o_{t+1})] \qquad (9)
\end{aligned}
$$

Since the agent has not experienced time step $t+1$ yet and has not received observations $o_{t+1}$, we take expectation over $o_{t+1}$ using the likelihood $p(o_{t+1}|s_{t+1})$ as (eq.7). In (eq.8), we approximate $p(o_{t+1}|s_{t+1})$ as $q(o_{t+1}|s_{t+1})$ and $p(s_{t+1}|o_{t+1})$ as $q(s_{t+1}|o_{t+1})$. According to the complete class theorem (Friston et al., 2012), any scalar rewards can be encoded as observation priors using $p(o) \propto \exp r(o)$ and the second term in (eq.9) becomes a goal-directed value. This observation prior $p(o_{t+1})$ can also be regarded as the probability of optimality variable $p(\mathcal{O}_{t+1} = 1|o_{t+1})$, where the binary optimality variable $\mathcal{O}_{t+1} = 1$ denotes that time step $t+1$ is optimal and $\mathcal{O}_{t+1} = 0$ denotes that it is not optimal as introduced in the context of control as probabilistic inference(Levine, 2018). The first term in (eq.9) is called epistemic value that works as intrinsic motivation to further explore the world. Minimization of $-KL(q(s_{t+1}|o_{t+1})||q(s_{t+1}))$ means that the agent tries to experience as different states $s_{t+1}$ as possible given some imagined observations $o_{t+1}$. By minimizing the Expected Free Energy, the agent can infer the actions that explores the world and maximize rewards. This process is called 'active inference'.

## 3  DEEP FREE ENERGY NETWORK (FENET)

Perceptual learning deals with learning the generative model to best explain the agent's sensory inputs. If we think of not only observations but also actions given by the expert as a part of the sensory inputs, we can explain imitation leaning by using the concept of perceptual learning. Active inference deals with exploration and reward maximization, so it is compatible with reinforcement learning. By minimizing the same objective function, the Free Energy, we can deal with both imitation and RL.

In this section, we first introduce a policy prior for imitation and a policy posterior for RL. Second, we extend the Free Energy Principle to be able to accommodate these two policies in the same objective function, the Free Energy. Finally, we explain a detailed network architecture to implement the proposed method for solving image control tasks.

### 3.1  INTRODUCING A POLICY PRIOR AND A POLICY POSTERIOR

**Free Energy**    We extend the Free Energy from (eq.4) so that actions are a part of sensory inputs that the generative model tries to explain.

$$
F_t = KL(q(s_t)||p(o_t, s_t, a_t)) = KL(q(s_t)||p(o_t|s_t)p(a_t|s_t)p(s_t|s_{t-1}, a_{t-1})) \qquad (10)
$$

$$
= \mathbb{E}_{q(s_t)}[\ln \frac{q(s_t)}{p(o_t|s_t)p(a_t|s_t)p(s_t|s_{t-1}, a_{t-1})}] \qquad (11)
$$

$$
= \mathbb{E}_{q(s_t)}[-\ln p(o_t|s_t) - \ln p(a_t|s_t) + \ln q(s_t) - \ln p(s_t|s_{t-1}, a_{t-1})] \qquad (12)
$$

$$
= \mathbb{E}_{q(s_t)}[-\ln p(o_t|s_t) - \ln p(a_t|s_t)] + KL(q(s_t)||p(s_t|s_{t-1}, a_{t-1})) \qquad (13)
$$

We define $p(a_t|s_t)$ as a policy prior. When the agent observes expert trajectories, by minimizing $F_t$, the policy prior will be learned so that it can best explain the experts. Besides the policy prior, we introduce and define a policy posterior $q(a_t|s_t)$, which is the very policy that the agent samples from when interacting with its environments. We explain how to learn the policy posterior in the following.

**Expected Free Energy for imitation**    In a similar manner to active inference in Section 2.2, we think of the Expected Free Energy $G_{t+1}$ at the next time step $t+1$, but this time we take expectation over the policy posterior $q(a_t|s_t)$ because $G_{t+1}$ is a value expected under the next actions. Note that

in Section 2.2 $a_t$ was given as a certain value, but here $a_t$ is sampled from the policy posterior. We calculate the expected variational posterior at time step $t + 1$ as follows.

$$q(s_{t+1}) = \mathbb{E}_{q(s_t)q(a_t|s_t)}[p(s_{t+1}|s_t, a_t)] \tag{14}$$

$$q(o_{t+1}, s_{t+1}, a_{t+1}) = \mathbb{E}_{q(s_{t+1})}[p(o_{t+1}|s_{t+1})q(a_{t+1}|s_{t+1})] \tag{15}$$

We extend the Expected Free Energy from (eq.12) so that the variational posterior makes inference on actions as follows.

$$
\begin{aligned}
G_{t+1}^{IL} &= \mathbb{E}_{q(o_{t+1}, s_{t+1}, a_{t+1})}[-\ln p(o_{t+1}|s_{t+1}) - \ln p(a_{t+1}|s_{t+1}) + \ln q(s_{t+1}, a_{t+1}) \\
&\quad - \ln p(s_{t+1}|s_t, a_t)] \\
&= \mathbb{E}_{q(o_{t+1}, s_{t+1}, a_{t+1})}[-\ln p(o_{t+1}|s_{t+1}) - \ln p(a_{t+1}|s_{t+1}) + \ln q(a_{t+1}|s_{t+1})] \\
&\quad + KL(q(s_{t+1})||p(s_{t+1}|s_t, a_t)) \\
&= \mathbb{E}_{q(o_{t+1}, s_{t+1})}[-\ln p(o_{t+1}|s_{t+1}) + KL(q(a_{t+1}|s_{t+1})||p(a_{t+1}|s_{t+1}))] \\
&\quad + KL(q(s_{t+1})||p(s_{t+1}|s_t, a_t)) \\
&= \mathbb{E}_{q(o_{t+1}, s_{t+1})}[-\ln p(o_{t+1}|s_{t+1}) + KL(q(a_{t+1}|s_{t+1})||p(a_{t+1}|s_{t+1}))] + 0 \\
&= \mathbb{E}_{q(s_{t+1})}[\mathcal{H}[p(o_{t+1}|s_{t+1})] + KL(q(a_{t+1}|s_{t+1})||p(a_{t+1}|s_{t+1}))]
\end{aligned}
$$

$$\tag{16}$$
$$\tag{17}$$
$$\tag{18}$$
$$\tag{19}$$
$$\tag{20}$$

In (eq.20), the first term is the entropy of the observation likelihood, and the second term is the KL divergence between the policy prior and the policy posterior. By minimizing $G_{t+1}^{IL}$, the agent learns the policy posterior so that it matches the policy prior which has been learned through minimizing $F_t$ to encode the experts' behavior.

**Expected Free Energy for RL**    We can get the Expected Free Energy in a different way that has a reward component $r(o_{t+1})$ leading to the policy posterior maximizing rewards. We extend the Expected Free Energy from (eq.8) so that the variational posterior makes inference on actions as follows.

$$
\begin{aligned}
G_{t+1}^{RL} &= \mathbb{E}_{q(o_{t+1}, s_{t+1}, a_{t+1})}[\ln q(s_{t+1}, a_{t+1}) \\
&\quad - \ln p(a_{t+1}|s_{t+1}) - \ln q(s_{t+1}|o_{t+1}) - \ln p(o_{t+1})] \\
&= \mathbb{E}_{q(o_{t+1}, s_{t+1})}[\ln q(s_{t+1}) - \ln q(s_{t+1}|o_{t+1}) \\
&\quad + KL(q(a_{t+1}|s_{t+1})||p(a_{t+1}|s_{t+1})) - \ln p(o_{t+1})] \\
&= \mathbb{E}_{q(o_{t+1})}[-KL(q(s_{t+1}|o_{t+1})||q(s_{t+1})) - \ln p(o_{t+1})] \\
&\quad + \mathbb{E}_{q(s_{t+1})}[KL(q(a_{t+1}|s_{t+1})||p(a_{t+1}|s_{t+1}))] \\
&\approx \mathbb{E}_{q(o_{t+1})}[-KL(q(s_{t+1}|o_{t+1})||q(s_{t+1})) - r(o_{t+1})] \\
&\quad + \mathbb{E}_{q(s_{t+1})}[KL(q(a_{t+1}|s_{t+1})||p(a_{t+1}|s_{t+1}))]
\end{aligned}
$$

$$\tag{21}$$
$$\tag{22}$$
$$\tag{23}$$
$$\tag{24}$$

In a similar manner to active inference in Section 2.2, we use $p(o) \propto \exp r(o)$ in (eq.24). The first KL term is the epistemic value that lets the agent explore the world, the second term is the expected reward under the action sampled from the policy posterior, and the last KL term is the KL divergence between the policy prior and the policy posterior. The last KL term can be written as follows (eq.25), meaning that minimizing this term leads to maximizing the entropy of the policy posterior at the same time the policy posterior tries to match the policy prior. Thus, the expected free energy can be regarded as one of entropy maximizing RL methods.

$$KL(q(a_{t+1}|s_{t+1})||p(a_{t+1}|s_{t+1})) = -\mathcal{H}[q(a_{t+1}|s_{t+1})] - \mathbb{E}_{q(a_{t+1}|s_{t+1})}[\ln p(a_{t+1}|s_{t+1})] \tag{25}$$

Note that $q(o_{t+1})$ in (eq.24) can be calculated as follows.

$$q(o_{t+1}) = \mathbb{E}_{q(s_{t+1})}[p(o_{t+1}|s_{t+1})] \tag{26}$$

By minimizing $G_{t+1}^{RL}$, the agent learns the policy posterior so that it explores the world and maximizes the reward as long as it does not deviate too much from the policy prior which has encoded experts' behavior through minimizing $F_t$.

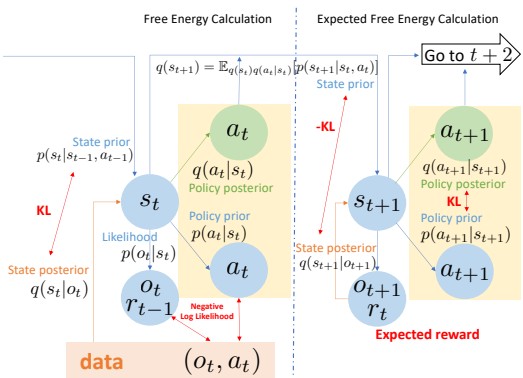

Figure 1: Deep Free Energy Network (FENet) calculation process. The left side shows how to calculate the Free Energy using data at hand. The right side shows how to calculate the Expected Free Energy for RL with latent imagination.

### 3.2 IMITATION AND RL OBJECTIVES

To account for the long term future, the agent has to calculate the Expected Free Energy at $t + 1$ to $\infty$.

$$\mathcal{F} = F_t + \sum_{\tau=t+1}^{\infty} \gamma^{\tau-t-1} G_\tau \qquad (27)$$

We define this curly $\mathcal{F}$ to be the objective that the Deep Free Energy Network should minimize. Note that $\gamma$ is a discount factor as in the case of general RL algorithms. As it is impossible to sum over infinity time steps, we introduce an Expected Free Energy Value function $V(s_{t+1})$ to estimate the cumulative Expected Free Energy. Similarly to the case of Temporal Difference learning of Deep Q Network (Mnih et al., 2013), we use a target network $V_{targ}(s_{t+2})$ to stabilize the learning process and define the loss for the value function as follows.

$$\mathcal{L} = ||G_{t+1} + \gamma V_{targ}(s_{t+2}) - V(s_{t+1})||^2 \qquad (28)$$

We made a design choice that the agent uses the value function only for RL, and not for imitation. In imitation, we use only the real value of the Expected Free Energy $G_{t+1}$ at the next time step $t+1$. This is because imitation learning can be achieved without long term prediction as the agent is given the experts' all time series data available. On the other hand, in RL, using the value function to predict rewards in the long-term future is essential to avoid a local minimum and achieve the desired goal.

In conclusion, the objective functions of Deep Free Energy Network (FENet) for a data sequence $(o_t, a_t, r_t, o_{t+1})$ are as follows.

$$\mathcal{F}_{IL} = F_t + G_{t+1}^{IL} \qquad (29)$$

$$\mathcal{F}_{RL} = F_t + G_{t+1}^{RL} + \gamma V_{\omega_{targ}}(s_{t+2}) \qquad (30)$$

$$\mathcal{L} = ||G_{t+1}^{RL} + \gamma V_{targ}(s_{t+2}) - V(s_{t+1})||^2 \qquad (31)$$

The overall Free Energy calculation process is shown in Figure 1.

### 3.3 NETWORK ARCHITECTURE AND CALCULATION

For implementation, we made a design choice to use Recurrent State Space Model (Hafner et al., 2019b), a latent dynamics model with both deterministic and stochastic components. In this model, the hidden states $s_t$ are split into two parts: stochastic hidden states $s_t$ and deterministic hidden states $h_t$. The deterministic transition of $h_t$ is modeled using Recurrent Neural Networks (RNN) $f$ as follows.

$$h_t = f(h_{t-1}, s_{t-1}, a_{t-1}) \qquad (32)$$

We model the probabilities in Deep Free Energy Networks as follows.

| | | |
|---|---|---|
| State prior | $p_\theta(s_t|h_t)$ | (33) |
| Observation likelihood | $p_\theta(o_t|s_t, h_t)$ | (34) |
| Reward likelihood | $p_\theta(r_{t-1}|s_t, h_t)$ | (35) |
| State posterior | $q_\phi(s_t|h_t, o_t)$ | (36) |
| Policy prior | $p_\theta(a_t|s_t, h_t)$ | (37) |
| Policy posterior | $q_\psi(a_t|s_t, h_t)$ | (38) |
| Value network | $V_\omega(s_t)$ | (39) |
| Target Value Network | $V_{\omega_{targ}}(s_t)$ | (40) |

We model these probabilities as feedforward Neural Networks that output the mean and standard deviation of the random variables according to the Gaussian distribution. The parameters $\theta, \phi, \psi, \omega$ are network parameters to be learned. Using the network parameters, the objective loss functions can be written as follows.

$$\mathcal{F}_{IL} = F_t + G_{t+1}^{IL} \tag{41}$$

$$\mathcal{F}_{RL} = F_t + G_{t+1}^{RL} + \gamma V_{\omega_{targ}}(s_{t+2}) \tag{42}$$

$$\mathcal{L} = ||G_{t+1}^{RL} + \gamma V_{\omega_{targ}}(s_{t+2}) - V_\omega(s_{t+1})||^2 \tag{43}$$

when

$$F_t = \mathbb{E}_{q_\phi(s_t|h_t,o_t)}[-\ln p_\theta(o_t|s_t, h_t) - \ln p_\theta(a_t|s_t, h_t)] + KL(q_\phi(s_t|h_t, o_t)||p_\theta(s_t|h_t)) \tag{44}$$

$$G_{t+1}^{IL} = \mathbb{E}_{q(s_{t+1})}[\mathcal{H}[p_\theta(o_{t+1}|s_{t+1}, h_{t+1})] + KL(q_\psi(a_{t+1}|s_{t+1}, h_{t+1})||p_\theta(a_{t+1}|s_{t+1}, h_{t+1}))] \tag{45}$$

$$G_{t+1}^{RL} = \mathbb{E}_{q(o_{t+1})}[-KL(q_\phi(s_{t+1}|h_{t+1}, o_{t+1})||q(s_{t+1})) - p_\theta(r_t|s_{t+1}, h_{t+1})]$$
$$+ \mathbb{E}_{q(s_{t+1})}[KL(q_\psi(a_{t+1}|s_{t+1}, h_{t+1})||p_\theta(a_{t+1}|s_{t+1}, h_{t+1}))] + \gamma V_{\omega_{targ}}(s_{t+2}) \tag{46}$$

$$q(s_{t+1}) = \mathbb{E}_{q_\phi(s_t|h_t,o_t)q_\psi(a_t|s_t,h_t)}[p_\theta(s_{t+1}|h_{t+1})] \tag{47}$$

$$q(o_{t+1}) = \mathbb{E}_{q(s_{t+1})}[p_\theta(o_{t+1}|s_{t+1}, h_{t+1})] \tag{48}$$

Algorithm 1 in Appendix shows overall calculations using these losses. The agent minimizes $\mathcal{F}_{IL}$ for expert data $\mathcal{D}_E$ and the agent minimizes $\mathcal{F}_{RL}$ for agent data $\mathcal{D}_A$ that the agent collects on its own.

## 4 EXPERIMENTS

We evaluate FENet on three continuous control tasks from images. We compare our model with model-based RL and model-based imitation RL in dense and sparse reward setting when optimal expert is available. Then we compare our model with imitation learning methods when only suboptimal experts are available. Finally, we investigate the merits of combining imitation and RL as an ablation study.

**Control tasks**  We used Cheetah-run, Walker-walk, and Quadruped-walk tasks, image-based continuous control tasks of DeepMind Control Suite (Tassa et al., 2018) shown in Figure 6. The agent gets rewards ranging from $0$ to $1$. Quadruped-walk is the most difficult as it has more action dimensions than the others. Walker-walk is more challenging than Cheehtah-run because an agent first has to stand up and then walk, meaning that the agent easily falls down on the ground, which is difficult to predict. The episode length is 1000 steps starting from randomized initial states. We use action repeat $R = 4$ for the Cheetah-run task, and $R = 2$ for the Walker-walk task and the Quadruped-walk task.

### 4.1 PERFORMANCE IN STANDARD VISUAL CONTROL TASKS

We compare the performance of FENet to PlaNet (RL) and "PlaNet with demonstrations" (imitation RL) in standard visual control tasks mentioned above. We use PlaNet as a baseline method because PlaNet is one of the most basic methods using Recurrent State Space Model, on top of which

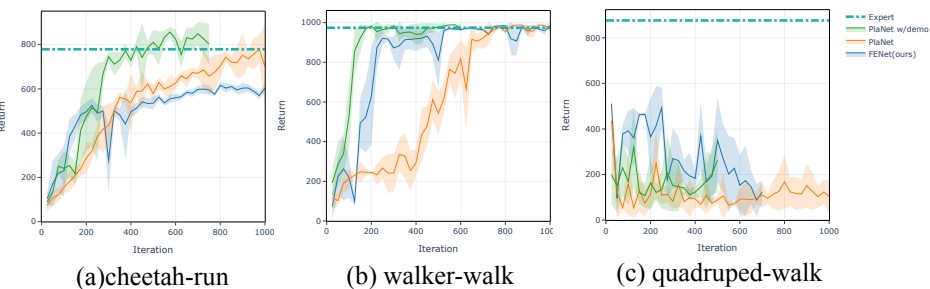

Figure 2: Comparison of FENet to PlaNet and "PlaNet with demonstrations". Plots show test performance over learning iterations. The lines show means and the areas show standard deviations over 10 trajectories.

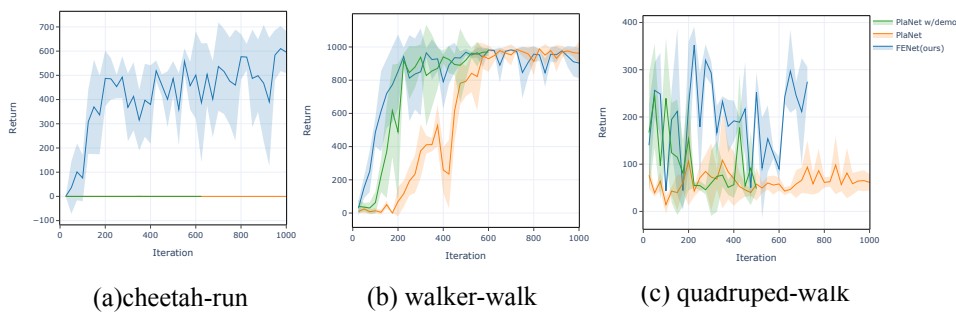

Figure 3: Comparison of FENet to PlaNet and "PlaNet with demonstrations" in sparse-reward settings, where agents do not get rewards less than 0.5. Plots show test performance over learning iterations. FENet substantially outperforms PlaNet. The lines show means and the areas show standard deviations over 10 trajectories.

we build our model. As FENet uses expert data, we create "PlaNet with demonstrations" for fair comparison. This variant of PlaNet has an additional experience replay pre-populated with expert trajectories and minimize a loss calculated from the expert data in addition to PlaNet's original loss.

Figure 2 shows that "PlaNet with demonstrations" is always better than PlaNet and that FENet is ranked higher as the difficulty of tasks gets higher. In Cheetah-run, FENet gives competitive performance with PlaNet. In Walker-walk, FENet and "PlaNet with demonstrations" are almost competitive, both of which are substantially better than PlaNet thanks to expert knowledge being leveraged to increase sample efficiency. In Quadruped-walk, FENet is slightly better than the other two baselines.

## 4.2 PERFORMANCE IN SPARSE-REWARD VISUAL CONTROL TASKS

In real-world robot learning, it is demanding to craft a dense reward function to lead robots to desired behaviors. It would be helpful if an agent could acquire desired behaviors simply by giving sparse signals. We compare the performance of FENet to PlaNet and "PlaNet with demonstrations" in sparse-reward settings, where agents do not get rewards less than 0.5 per time step (Note that in the original implementation of Cheetah-run, Walker-walk and Quadruped-walk, agents get rewards ranging from 0 to 1 per time step). Figure 3 shows that FENet outperforms PlaNet and "PlaNet with demonstrations" in all three tasks. In Cheetah-run, PlaNet and "PlaNet with demonstrations" are not able to get even a single reward.

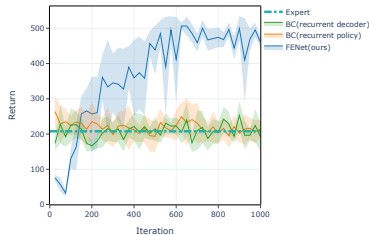

Figure 4: Comparison of FENet to imitation learning methods when only suboptimal experts are available in Cheetah-run. Plots show test performance over learning iterations. Behavioral Cloning imitation methods cannot surpass the suboptimal expert's return which FENet successfully surpasses. The lines show means and the areas show standard deviations over 10 trajectories.

### 4.3 PERFORMANCE WITH SUBOPTIMAL EXPERTS

In real-world robot learning, expert trajectories are often given by human experts. It is natural to assume that expert trajectories are suboptimal and that there remains much room for improvement. We compare the performance of FENet to Behavioral Cloning imitation methods. We use two types of networks for behavioral cloning methods: recurrent policy and recurrent decoder policy. The recurrent policy $\pi_R(a_t|o_t)$ is neural networks with one gated recurrent unit cell and three dense layers. The recurrent decoder policy $\pi_R(a_t, o_{t+1}|o_t)$ is neural networks with one gated recurrent unit cell and four dense layers and deconvolution layers as in the decoder of PlaNet. Both networks does not get raw pixel observations but take observations encoded by the same convolutional encoder as PlaNet's.

Figure 4 shows that while imitation methods overfit to the expert and cannot surpass the suboptimal expert performance, FENet is able to substantially surpass the suboptimal expert's performance.

### 4.4 LEARNING STRATEGIES

Figure 5 compares learning strategies of FENet in Cheetah-run and Walker-walk (ablation study). 'Imitation RL' is the default FENet agent that does imitation learning and RL at the same time, minimizing $\mathcal{F}_{IL} + \mathcal{F}_{RL}$. 'Imitation-pretrained RL' is an agent that first learns the model only with imitation (minimizing $\mathcal{F}_{IL}$) and then does RL using the pre-trained model (minimizing $\mathcal{F}_{RL}$). 'RL only' is an agent that does RL only, minimizing $\mathcal{F}_{RL}$. 'Imitation only' is an agent that does imitation only, minimizing $\mathcal{F}_{IL}$. While 'imitation only' gives the best performance and 'imitation RL' gives the second best in Cheetah-run, 'imitation RL' gives the best performance and 'imitation only' gives the worst performance in Walker-walk. We could say 'imitation RL' is the most robust to the properties of tasks.

## 5 RELATED WORK

**Active Inference** Friston, who first proposed Active Inference, has evaluated the performance in simple control tasks and a low-dimensional maze (Friston et al., 2012; 2015). Ueltzhoffer implemented Active Inference with Deep Neural Networks and evaluated the performance in a simple control task (Ueltzhöffer, 2018). Millidge proposed a Deep Active Inference framework with value functions to estimate the correct Free Energy and succeeded in solving Gym environments (Millidge, 2019). Our approach extends Deep Active Inference to combine imitation and RL, solving more challenging tasks.

**RL from demonstration** Reinforced Imitation Learning succeeds in reducing sample complexity by using imitation as pre-training before RL (Pfeiffer et al., 2018). Adding demonstrations into a replay buffer of off policy RL methods also demonstrates high sample efficiency (Vecerik et al., 2017; Nair et al., 2018; Paine et al., 2019). Demo Augmented Policy Gradient mixes the policy gradient with a behavioral cloning gradient (Rajeswaran* et al., 2018). Deep Q-learning from Demonstrations (DQfD) not only use demonstrations for pre-training but also calculates gradients

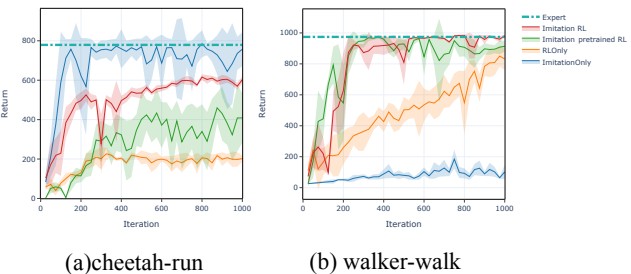

(a)cheetah-run        (b) walker-walk

Figure 5: Comparison of FENet (imitation RL) to other learning strategies (ablation studies: imitation-pretrained RL, RL only, and imitation only with FENets). Plots show test performance over learning iterations. The lines show means and the areas show standard deviations over 10 trajectories.

from demonstrations and environment interaction data (Hester et al., 2018). Truncated HORizon Policy Search uses demonstrations to shape rewards so that subsequent planning can achieve superior performance to RL even when experts are suboptimal (Sun et al., 2018). Soft Q Imitation Learning gives rewards that encourage the agent to return to demonstrated states in order to avoid policy collapse (Reddy et al., 2019). Our approach is similar to DQfD in terms of mixing gradients calculated from demonstrations and from environment interaction data. One key difference is that FENet concurrently learns the generative model of the world so that it can be robust to wider environment properties.

**Control with latent dynamics model**    World Models acquire latent spaces and dynamics over the spaces separately, and evolve simple linear controllers to solve visual control tasks (Ha & Schmidhuber, 2018). PlaNet learns Recurrent State Space Model and does planning with Model Predictive Control at test phase (Hafner et al., 2019b). Dreamer, which is recently built upon PlaNet, has a policy for latent imagination and achieved higher performance than PlaNet (Hafner et al., 2019a). Our approach also uses Recurrent State Space Model to describe variational inference, and we are the first to combine imitation and RL over latent dynamics models to the best of our knowledge.

## 6    CONCLUSION

We present FENet, an agent that combines Imitation Learning and Reinforcement Learning using Free Energy objectives. For this, we theoretically extend the Free Energy Principle and introduce a policy prior that encodes experts' behaviors and a policy posterior that learns to maximize expected rewards without deviating too much from the policy prior. FENet outperforms model-based RL and imitation RL especially in visual control tasks with sparse rewards and FENet also outperforms suboptimal experts' performance unlike Behavioral cloning. Strong potentials in sparse environment with suboptimal experts are important factors for real-world robot learning.

Directions for future work include learning the balance between imitation and RL, i.e. Free Energy and Expected Free Energy so that the agent can select the best approach to solve its confronting tasks by monitoring the value of Free Energy. It is also important to evaluate FENet in real-world robotics tasks to show that our method is effective in more realistic settings that truly appear in the real world.

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

## A APPENDIX

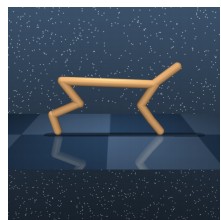 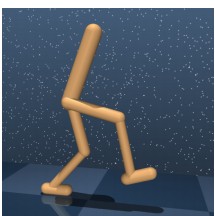 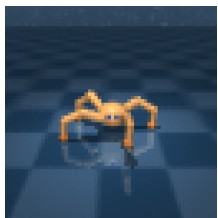

(a) Cheetah-run          (b) Walker-walk          (c) Quadruped-walk

Figure 6: Image-based control tasks used in our experiments.

### A.1 FENET ALGORITHM

See Algorithm 1.

### A.2 IMPLEMENTATION

To stabilize the learning process, we adopt burn-in, a technique to recover initial states of RNN's hidden variables $h_t$ (Kapturowski et al., 2019). As shown in Algorithm 1, the agent calculates the Free Energy with mini batches sampled from the expert or agent experience replay buffer $\mathcal{D}$, which means that $h_t$ is initialized randomly in every mini batch calculation. Since the Free Energy heavily depends on $h_t$, it is crucial to estimate the accurate hidden states. We set a burn-in period when a portion of the replay sequence is used only for unrolling the networks to produce initial states. After the period, we update the networks only on the remaining part of the sequence.

We use PyTorch (Paszke et al., 2017) to write neural networks and run experiments using NVIDIA GeForce GTX 1080 Ti / RTX 2080 Ti / Tesla V100 GPU (1 GPU per experiment). The training time for our FENet implementation is about 24 hours on the Control Suite environment. As for the hyper parameters, we use the convolutional encoder and decoder networks from (Ha & Schmidhuber, 2018) and Recurrent State Space Model from (Hafner et al., 2019b) and implement all other functions as three dense layers of size 200 with ReLU activations (Nair & Hinton, 2010). We made a design choice to make the policy prior, the policy posterior, and the observation likelihood, the reward likelihood deterministic functions while the state prior and the state posterior are stochastic. We use the batch size $B = 25$ for 'imitation RL' with FENet, and $B = 50$ for other types and baseline methods. We use the chunk length $L = 50$, the burn-in period 20. We use seed episodes $S = 40$, expert episodes $N = 10000$ trained with PlaNet (Hafner et al., 2019b), collect interval $C = 100$ and action exploration noise Normal(0, 0.3). We use the discount factor $\gamma = 0.99$ and the

---

**Algorithm 1** Deep Free Energy Network (FENet)

**Input:**
Seed episodes $S$
Collect interval $C$
Batch size $B$
Chunk length $L$
Expert episodes $N$
Target smoothing rate $\rho$
Learning rate $\alpha$
State prior $p_\theta(s_t|h_t)$
State posterior $q_\phi(s_t|h_t, o_t)$
Policy prior $p_\theta(a_t|s_t, h_t)$
Policy posterior $q_\psi(a_t|s_t, h_t)$
Likelihood $p_\theta(o_t|s_t, h_t), p_\theta(r_{t-1}|s_t, h_t)$
Value function $V_\omega(s_t)$
Target value function $V_{\omega_{targ}}(s_t)$

Initialize expert dataset $\mathcal{D}_E$ with $N$ expert trajectories
Initialize agent dataset $\mathcal{D}_A$ with $S$ random episodes
Initialize neural network parameters $\theta, \phi, \psi, \omega$ randomly
**while** not converged **do**
    **for** update step $c = 1..C$ **do**
        `// Imitation Learning`
        Draw expert data $\{(o_t, a_t, r_t, o_{t+1})_{t=k}^{k+L}\}_{i=1}^B \sim \mathcal{D}_E$
        Compute Free Energy $\mathcal{F}_{IL}$ from equation 41
        `// Reinforcement Learning`
        Draw agent data $\{(o_t, a_t, r_t, o_{t+1})_{t=k}^{k+L}\}_{i=1}^B \sim \mathcal{D}_A$
        Compute Free Energy $\mathcal{F}_{RL}$ from equation 42
        Compute $V$ function's Loss $\mathcal{L}$ from equation 43
        `// Update parameters`
        $\theta \leftarrow \theta - \alpha\nabla_\theta(\mathcal{F}_{IL} + \mathcal{F}_{RL})$
        $\phi \leftarrow \phi - \alpha\nabla_\phi(\mathcal{F}_{IL} + \mathcal{F}_{RL})$
        $\psi \leftarrow \psi - \alpha\nabla_\psi(\mathcal{F}_{IL} + \mathcal{F}_{RL})$
        $\omega \leftarrow \omega - \alpha\nabla_\omega\mathcal{L}$
        $\omega_{targ} \leftarrow \rho\omega_{targ} + (1 - \rho)\omega$
    **end for**
    `// Environment interaction`
    $o_1 \leftarrow$ `env.reset()`
    **for** time step $t = 1..T$ **do**
        Infer hidden states $s_t \leftarrow q_\phi(s_t|h_t, o_t)$
        Calculate actions $a_t \leftarrow q_\psi(a_t|s_t, h_t)$
        Add exploration noise to actions
        $r_t, o_{t+1} \leftarrow$ `env.step` $(a_t)$
    **end for**
    $\mathcal{D}_A \leftarrow \mathcal{D}_A \cup \{(o_t, a_t, r_t, o_{t+1})_{t=1}^T\}$
**end while**

---

target smoothing rate $\rho = 0.01$. We use Adam (Kingma & Ba, 2014) with learning rates $\alpha = 10^{-3}$ and scale down gradient norms that exceed 1000. We scale the reward-related loss by 100, the policy-prior-related loss by 10. We clip KL loss between the hidden states below 3 free nats and clip KL loss between the policies below 0.6.

