# OpenReview forum: "Combining Imitation and Reinforcement Learning with Free Energy Principle"
_ICLR.cc/2021/Conference — Reject_

### Official Review · AnonReviewer1 · 2020-10-20
**My main concerns are related to the clarity of the paper**

**Rating:** 4
**Confidence:** 4

**Review:**

This paper aims at merging Imitation Learning (IL) and Reinforcement Learning (RL) from high dimensional sensory inputs so that the agent can make use of expert trajectories, even when the experts are suboptimal. This paper aims at addressing this by theoretically using the "Free Energy Principle", which the authors define in the abstract as a unified brain theory that explains perception.

The paper tackles an important problem and develops many theoretical functionals. The approach is then tested on a few benchmarks from the DeepMind Control Suite where the approach is reported to work well.

My main concerns are related to the clarity of the paper, which does not allow me to understand some key parts.

It is unclear what are the minimized loss functions: it is mentioned that equations 27-29 sum up all objective functions. Since F_t and G_{t+1}^{RL} are appearing twice, does it mean that those losses have a two times more important contribution than for instance G_{t+1}^{IL}? In addition how exactly are equations 27-29 derived from the other equations?

Notations are not always consistent:
- F_{IL} F_{RL} are sometimes with curly F sometimes not (equations 27-29 and the few lines that follow Figure 1 page 5).
- First line equation 7 and first line equation 10: Why is there a subscript for F and not G? In other parts of the paper, a subscript is used for G as well.

Some sentences are unclear:
- "in RL, using the value function to predict rewards in the long-term future is essential to avoid a local minimum and achieve the desired goal." What kind of local minimum does that refer to?

A few typos:
- "Then We (...)"
- "the agent easily fall down"

---

> ### Author Response · Authors · 2020-11-23
> **Response to Reviewer 1**
>
> We thank the reviewer for the comments. We have revised the paper according to the suggestions and would like to clarify several things:
>
> ・It is unclear what are the minimized loss functions
> > I have updated the paper, and please find eq. 41-43 and Algorithm 1 in Appendix. If you look at Algorithm 1, you can see that F_IL + F_RL are minimized along parameters theta, phi and psi. L is simply a loss along parameter omega for learning a value function which is used to calculate F_RL as in eq. 42. Thus, the contributions of F_IL (incl. G_{t+1}^{IL}) and F_RL (incl. G_{t+1}^{RL}) are the same. F_IL and F_RL are derived from eq. 27 (sum of free energy at future time steps).
>
> ・Notations are not always consistent:
> > I’ve updated all F_IL, F_RL to curly F. I’ve added a subscript for all F and G.
>
> ・Unclear sentence "in RL, using the value function to predict rewards in the long-term future is essential to avoid a local minimum and achieve the desired goal."
> > What I mean by local minimum here is the case where an agent cannot achieve the global minimum of the loss by pursuing rewards in a short time horizon. For example, in "mountain car problem", the agent needs to temporarily take negative-reward actions (not local minimum) to gain momentum to go up the hill, which leads to higher rewards in the end.
>
> ・Typos
> > I fixed the typos. Thanks for pointing these out.

---

> > ### Comment · AnonReviewer1 · 2020-11-24
> > **still not clear to me**
> >
> > Thanks for the reply. I still can't understand exactly what are the functionals that are minimized. Is it all equations 41+42+43 ?  And where do the $\theta, \phi$ and $\psi$ actually appear? It would be more readable if they appear with a notation such as $f(\cdot; \theta)$.
> >
> > In other words, the following part in your reply is still not understandable to me: "you can see that F_IL + F_RL are minimized along parameters theta, phi and psi. L is simply a loss along parameter omega for learning a value function which is used to calculate F_RL as in eq. 42."
> >
> >  In addition, the motivation to justify equations 41-43 is also unclear. Unfortunately, at this point I keep my score unchanged.

---

> > > ### Author Response · Authors · 2020-11-25
> > > **Let me explain in details**
> > >
> > > Thanks for the reply. Let me explain more in details.
> > >
> > > Q1. What are the functionals that are minimized? Where do the theta, phi, and psi actually appear?
> > >
> > > A1.
> > > Eq.41, 42, 43 are all minimized, but updated parameters are different. Let me explain this step by step. Please look at section 3.3 while you read the followings (especially eq. 33-40).
> > >
> > > We model a probability with Deep Neural Networks that output the mean and standard deviation of the probability and regard the output as Gaussian distribution. For example, policy posterior is modeled as a network that takes s_t and h_t as inputs and calculates through several hidden layers and outputs the Gaussian distribution of a_t.
> > >
> > > Theta is a group/set of parameters consisting the probabilities of state prior, observation/reward likelihood and policy prior.
> > > Phi is a group of parameters consisting the probability of state posterior.
> > > Psi is a group of parameters consisting the probability of policy posterior.
> > > Omega is a group of parameters consisting the probability of value network.
> > > For example, equation 38 can also be written as q(a_t|s_t, h_t; psi) as you suggested.
> > >
> > > Group of parameters theta, phi and psi are updated (gradient descent) to minimize Eq. 41+42.  Group of parameters omega is updated to minimize Eq 43.
> > >
> > > In summary, eq.41 + 42 (F_IL + F_RL) is minimized to update a set of parameters theta, phi and psi. Eq. 43 (L) is minimized to update a set of parameters omega.
> > >
> > >
> > > Q2. What are the motivation to justify equation 41-43?
> > >
> > > A2.
> > > Eq. 41 and 42 are the free energy objectives and eq. 43 is a loss for the value function (approximator of the sum of expected free energy at time step t+2 ~ infinity).
> > >
> > > About eq. 41 and 42:
> > > According to Free Energy Principle (please see section 2 or  reference of Friston's work for more details), by minimizing the free energy, the agent can learn variational (state) posterior, model (prior and likelihood), and action (policy). If you look at eq.7, you can understand that the expected free energy is derived by imagining 1  time step ahead. Through the derivation in section 3.1, we (mathematically) extended the free energy and the expected free energy to be readily used for imitation and RL. If you look at eq. 27, you can understand that the curly F is a sum of free energy and expected free energies (t+1 to infinitiy). In other words, the curly F explains now (t) and future (t+1 to infinity; the agent has not experienced this yet). Eq. 29 and 30 are both derived from eq. 27. The reason why we separate these two is that we have different formulation (G^IL and G^RL) depending on IL or RL, both of which are from the same mathematical amount (imagining 1 time step ahead from F from eq. 10). Eq. 41 and 42 are the same as eq. 29 and 30 respectively.
> > >
> > > About eq. 43:
> > > This is not a free energy objective. This loss is for learning a value function which tries to approximate G^{t+2} + … G^{infinity}. The reason why we introduce this is that it is impossible to add up the amounts until infinity. To learn the value function, we use the exactly same way as temporal difference learning of Deep Q Network. Please see this prior work for the validity.
> > >
> > >
> > > Thank you very much for your comments! We will update the paper for the camera ready version so that every reader can understand.

---

### Official Review · AnonReviewer4 · 2020-10-29
**An interesting paper shows theoretical framework to explain imitation learning and RL with free energy principle**

**Rating:** 6
**Confidence:** 5

**Review:**

This manuscript develops RL algorithms that can learn reward from expert data and even exceed the expert performance. It introduces free energy framework that combines ideas of imitation learning and reinforcement learning in a Bayesian probabilistic way. The methodology is well written. It is interesting that they find the algorithm can deal with both imitation learning and RL by minimizing the same Free energy objective function.  The expert data provides policy prior. Then, the policy posterior will try to match the policy prior as the imitation learning, and at the same time, it is motivated to maximize the reward and explore the environment by maximizing the KL divergence between the state posterior and the state posterior given imaged observations.

With the proposed objective function, the algorithm applies existing recurrent state space model to solve the problem. Though it applies existing model, it would enhance the paper if it can show further analysis of the algorithm in this context, particularly from perspective of imitation learning. And the paper may need to enhance the empirical study by testing more problems and provide convincing analysis.

The math notations could be more consistent, as in Section 3 it widely uses variables like q(s_{\tau}), q(s_{\tau}|o_{\tau}), p(a_{t}|s_{t}), but in Appendix, there is no  q(s_{\tau}), it defines the variables like q_{\phi}(s_{t}|h_{t}, o_{t}), p_{\theta}(a_{t}|s_{t}, h_{t}) etc.

It may lack details on the the gradient of energy function. In Algorithm 1, there are important steps to update the parameters, it is not straightforward to find how to get the gradient from the functions 27 - 29.

In section 3.2, it may need further clarification on why imitation learning doesn’t need longer term objective. The assumption seems not straightforward, as usually the agent mimics the expert behavior to guarantee its expected long term reward is similar as the expert.

Though the abstract mentions the proposed algorithm can achieve better performance than standard IL or RL, there isn’t sufficient evidence. It may need theoretical analysis or rich empirical results. The experiments could study more simulation problems, and may compare with state-of-the-art inverse RL algorithms. Other questions related to experiments include:
- How to set up the expert data for  “PlaNet with demonstrations” in Section 4.2?
- Can experiments show the reward shaping benefit using virtualization to compare different reward estimations and the ground truth?
- The behavior cloning algorithms compared in Section 4.3 seem basic in the field, it may be worth comparing with “PlaNet with suboptimal expert data”, or other inverse RL algorithms.
- In Section 4.4, it may need to provide reference to the algorithms in study. Are they known algorithms, eg. which imitation RL algorithm is used,  which algorithm is the imitation pretrained RL (which algorithm is used to learn model and which RL is employed  after that), and which RL algorithm is used in ‘RL only’.

Some questions are written below.
- On page 2, it could make clear which parameters are optimized to minimize the Free Energy function of Eq. (4)
- In Eq. (8) when it replaces p(s_{\tau}|o_{\tau}) with q(s_{\tau}|o_{\tau}), it may be better to explain how to get the approximation and whether there is error bound.
- It would be interesting to know the convergence analysis of Algorithm 1, is it always convergent.
- Section 3.1 mentions we give a_t a prior in Section 2.2, but it seems there isn't a prior of a_t in section 2.2.
- It may be worth explaining q(s_{\tau}) in Eq. 14, is it equal to the variational posterior  as an approximation of p(s_{t+1}|o_{t+1})  on page 2.?
- It may need explanation on how to reach Eq.15 from Eq.12? Is it directly derived from KL(q(o_{\tau}, s_{\tau}, a_{\tau})||p(o_{\tau}, s_{\tau}, a_{\tau})?
- Figure 1 may lack good explanation, as the graph looks complicated with many math notations, not self-explanatory.
- The notation $s_{t}$ in section 3.3 may be confusing as $s_{t}$ refers to both hidden state and the stochastic part of it. In Figure 1, the deterministic transition only depends on h_t and a_t, while it depends on s_{t-1} too in  Eq.30.
- In quadruped-walk experiments, like Figure 3,  it seems the solution is not convergent yet. Why do FEnet and planet w/demo have less number of iterations?

---

> ### Author Response · Authors · 2020-11-23
> **Response to Reviewer 4**
>
> We thank the reviewer for the comments. We have revised the paper according to the suggestions and would like to clarify several things:
>
> ・Math notation consistency
> > I have updated the paper for the notations to be more consistent. It would be great if you could have a look at sections 2&3 and Algorithm 1 in Appendix.
>
> ・Gradient of loss functions
> > I have added more explanation in section 3.3 so that readers can understand the gradient / how the loss functions are minimized along which network parameters.
>
> ・Clarification on why Imitation does not need longer term objective
> > As the expert data is created as demonstrations that maximizes long-term return beforehand, what the agent has to do in imitation learning is simply to match its policy to demonstrations at each time step without looking ahead. In contrast, in RL, the agent has to look at future rewards.
>
> ・Evidence on achieving better performance
> > I have updated the abstract. The main contribution of this paper is to propose a FEP-based framework that can deal with imitation and RL with the same objectives. Proving the method to be SOTA is not the scope of this paper.
>
> ・Expert data for “PlaNet with demonstrations” in section 4.2
> > As expert demonstration data do not require reward information, we used the same experts from non-sparse environments. For Cheetah and Walker, we used PlaNet at final performance as the experts. For Quadruped, we trained Soft Actor Critic in classic env (not visual control) and I rendered the expert demonstrations and used them as the expert.
>
> ・Algorithm in section 4.4
> >This section is about ablation studies. I have updated the paper so that readers can understand what each method is about. For example, RL only means the agent only uses F_RL as an objective function, ignoring F_IL.
>
> ・Eq. 4 on page 2
> > As is written, F is minimized along the parameters of the variational posterior q(s_t). For all the objective functions of FENet, I have updated section 3.3 to show each objective function is minimized along which parameters.
>
> ・Section 3.1 mentions “we give a_t a prior in Section 2.2”
> > Not “a prior”, but “a priori”. As the expression was confusing, I have updated this part to be more straightforward.
>
> ・Eq. 14, is it equal to the variational posterior as an approximation of p(s_{t+1}|o_{t+1}) on page 2 ?
> > Yes
>
> ・Figure 1 may lack good explanation
> > I have updated Figure 1.
>
> ・Confusion of s_t
> > I have updated the paper so that  s_t means the stochastic part only in section 3.3 and Algorithm1 (the last part of method description). In anywhere before section 3.3, s_t always means the entire hidden state. I hope this flow is natural and readers will not get confused.

---

### Official Review · AnonReviewer2 · 2020-10-30
**Interesting idea, needs clearer communication and experimental section**

**Rating:** 5
**Confidence:** 2

**Review:**

This paper extends and explains how to apply the "free energy principle" and active inference to RL and imitation learning. They implement a neural network approximation of losses derived this way and test on some control tasks. Importantly the tasks focus on here are imitation + control tasks. That is, there is both a reward signal but also demonstration trajectories. The demonstrations may be suboptimal. The compare against PLaNet, a latent planning based approach.

It was difficult to evaluate this work well, as I found the approach difficult to follow. My primary concern about this paper is connecting it with other work and ensuring the terminology and approach is as clear as possible.

Firstly, in the initial derivation up to equation 9. By defining the observation prior $p(o) \propto \exp r(o)$ I believe this results in the same thing as "RL as inference" (e.g. [1] which states that "formulation proposed by Friston (2009) is similar to the maximum entropy approach outlined in this survey"). It would be good to make this connection clear for the reader. In particular, at least from my perspective, connecting to existing maximum entropy RL is helpful for understanding this work.

However, this leads to a concern of terminology. I realize the use of the term "observation" for $p(o)$ is to maintain the connection with the free energy formulation, but if the "observation" is really just the reward, then in the context of an RL paper this seems to lead to a lot of confusion if when you say "observation" you really just mean reward. In other places observation seems to mean the observation in a POMDP (e.g. in the algorithm, where reward is a separate variable).

Another source of confusion is using $s_t$ to refer to both the entire hidden state and just the stochastic part (3.3).

It would helpful to define the RL problem formally that is attempted to be solved here. It appears that is a partially observed MDP (which makes the use of "observation" even more confusing). It is not clear the the test environments are particularly partially observable, since the image reveals the full state of the world and these tasks have been solved with models that don't deal with partially observable states.

I think another view of this paper is that it is applying maximum entropy RL approach to a partially observed MDP, learning a demonstration policy, and then using this demonstration policy as a prior on an RL objective policy. Approaching this topic from a different perspective is helpful, but it would be useful to make connection with existing literature much clearer and state an alternative way of viewing the approach from a more standard RL point of view. Alternatively, if these statements are incorrect contrasting what is different would benefit the reader.

The experimental comparisons seem weak for a few reasons. There seems to be a conflation between model-based RL solutions and RL approaches dealing with partial observability. As far as I can understand the approach introduced here is the second, a state transition function is learned but this never used for planning, only for inferring the hidden state of the world. For this reason, the only comparison being with a model-based control approach seems lacking. It would be interesting to compare against other model-free approaches that incorporate demonstrations such as [3] (this does not deal with partial observability, but it's not clear these problems are partially observable) or [4] ([2] is an example of using a GAIL objective as in [4] + and a reward as is done here). Finally, it appears one of the significant benefits of the algorithm introduced here is to work well with partially observed MDPs, it would be good to construct environments which test this more clearly.

Overall, I think this work introduces interesting ideas. However, it needs to connect this with existing work such as RL as inference view of RL and make the approach clearer. The experimental section would benefit for more comparison with other methods and experiments on environments which are more straightforwardly partially observable.

[1] Levine, Sergey. "Reinforcement learning and control as probabilistic inference: Tutorial and review." arXiv preprint arXiv:1805.00909 (2018).

[2] Zhu, Yuke, et al. "Reinforcement and imitation learning for diverse visuomotor skills." arXiv preprint arXiv:1802.09564 (2018).

[3] Vecerik, Mel, et al. "Leveraging demonstrations for deep reinforcement learning on robotics problems with sparse rewards." arXiv preprint arXiv:1707.08817 (2017).

[4] Torabi, Faraz, Garrett Warnell, and Peter Stone. "Generative adversarial imitation from observation." arXiv preprint arXiv:1807.06158 (2018).

---

> ### Author Response · Authors · 2020-11-23
> **Response to Reviewer 2**
>
> We thank the reviewer for the comments. We have revised the paper according to the suggestions and would like to clarify several things:
>
> ・Connection with standard RL contexts
> > I have updated the paper. Connection with optimality variable in RL as inference is now mentioned in “Active Inference” in section 2.2. Connection with Entropy RL is now mentioned in “Expected Free Energy for RL” in section 3.1. Also, if I rephrase FENet in a more RL way, it would be “learning a demonstration policy, and then using this demonstration policy as a prior on an RL objective policy” while maximizing the entropy of the policy. The interesting thing about this paper is that both demonstration policy and RL objective policy are trained on the same loss of the Free Energy.
>
> ・Confusion of s_t to refer to both the entire hidden state and just the stochastic part
> > I have updated the paper so that  s_t means the stochastic part only in section 3.3 and Algorithm1 (the last part of method description). In anywhere before section 3.3, s_t always means the entire hidden state. I hope this flow is natural and readers will not get confused.
>
> ・Define the RL problem
> > This paper solves POMDP. PlaNet paper says individual image observations generally do not reveal the full state of the environment and uses visual control tasks of Cheetah and Walker as one of the POMDP environments. For example, in visual control tasks, velocity information can be regarded as non/partially observable. If you wanted to talk about occlusion, there is no occlusion in Cheetah/Walker, but still they can be regarded as POMDP environments.
>
> ・Experimental comparison with model-based RL
> > It is correct that as you said the state transition function of FENet is learned but never used for planning, only for inferring the hidden state of the world. Please note that PlaNet is a model-based RL using RSSM (PlaNet does Model Predictive Control). In this paper, FENet is compared with PlaNet as one example of model-based RL.

---

### Official Review · AnonReviewer3 · 2020-10-31

**Rating:** 5
**Confidence:** 2

**Review:**

This paper  introduces two different interpretations of free energy minimization as a form of behavior cloning and reinforcement learning.

Strength:
This approach seems to have significant gains on the environments evaluated.
The approach appears novel to my knowledge.

Weaknesses:
I found that I was confused by the presentation of section 3.1. In particular, I think the authors should clarify the difference between prior and posterior policies in both the RL and imitation learning setting as they appear to be different.

Why does equation 17 to 18 follow? Isn't the posterior policy different than the policy prior, leading to the likelihood to the next state this is distinct from the prior probability of the next state?

For equation 22 to 23, is the assumption that the likelihood of the next state is proportional to inverse exponentiated reward? I think the statement should be said in the text.

How does the approach compare with other approaches that encourage entropy in the policy? Such as something like soft Q learning? Or some type of curiosity?

Can this approach be evaluated on more realistic environments other than deepmind control?

Post Rebuttal Update:

Due to the remaining confusion among reviewers about the equations in the manuscript, I maintain my score.

---

> ### Author Response · Authors · 2020-11-23
> **Response to Reviewer 3**
>
> We thank the reviewer for the comments. We have revised the paper according to the suggestions and would like to clarify several things:
>
> ・The difference between prior and posterior policies in both the RL and imitation learning setting
> > As you can see in Algorithm 1 in Appendix, there are 2 phases (imitation and RL) in one iteration. In imitation learning, the policy prior is trained to match expert data (minimizing F) while the policy posterior is trained to match the policy prior (minimizing G^IL). In RL, the policy prior is trained to match agent data (minimizing F) while the policy posterior is trained to maximize reward and explore the world at the same time it tries to match the policy prior. As I have updated section 3, it would be great if you could read this part again.
>
> ・Equation 18 to 19 (eq. 17 to 18 in the first draft)
> > The policy posterior is different from the policy prior, but in this paper, only the policy posterior is used not only when the agent moves in real, but also when the agent makes inference as in Figure 1. Thus, likelihood to the next state and prior probability of the next state are identical.
>
> ・The assumption that the likelihood of the next state is proportional to inverse exponentiated reward.
> > It is correct. I have updated the paper to state this assumption in “Expected Free Energy for RL” in section 3.1.
>
> ・Connection with entropy-maximizing policy
> > FENet maximizes the entropy of the posterior policy. I have updated the paper to explain this connection in “Expected Free Energy for RL” in section 3.1.
>
> ・More realistic environments other than deepmind control
> > It is possible to try FENet on more realistic environments. As the purpose of this paper is to introduce how to theoretically extend Free Energy, more practical experiments are left for future work.

---

### Decision · Program_Chairs · 2021-01-07
**Final Decision**

**Decision:**

Reject

**Comment:**

This paper proposes a new algorithm that combines imitation learning and reinforcement learning, based on an extension of the free energy principal. The expert's demonstrations are encoded as a policy prior, and a posterior policy is inferred by maximizing expected rewards. While at a high-level this is a promising direction, all the reviewers found the paper difficult to follow and verify its claims. This mostly due to a use of unusual and non-conistent notations. The authors are advised to take into account the issues about clarity that the reviewers raised and improve the readability of their paper accordingly.